# Adaptation and Validation of the S-NutLit Scale to Assess Nutritional Literacy in the Peruvian Population

**DOI:** 10.3390/ijerph21060707

**Published:** 2024-05-30

**Authors:** Rony Francisco Chilón-Troncos, Elizabeth Emperatriz García-Salirrosas, Dany Yudet Millones-Liza, Miluska Villar-Guevara

**Affiliations:** 1Unidad de Ciencias Empresariales, Escuela de Posgrado, Universidad Peruana Unión, Lima 15102, Peru; ronnychilon@upeu.edu.pe (R.F.C.-T.); dannie@upeu.edu.pe (D.Y.M.-L.); 2Faculty of Management Science, Universidad Autónoma del Perú, Lima 15842, Peru; 3Escuela Profesional de Administración, Facultad de Ciencias Empresariales, Universidad Peruana Unión, Lima 15102, Peru; 4Escuela Profesional de Administración, Facultad de Ciencias Empresariales, Universidad Peruana Unión, Juliaca 21101, Peru; miluskavillar@upeu.edu.pe

**Keywords:** public health, nutritional literacy, healthy eating, scale

## Abstract

Maintaining good dietary practices is a factor that allows a better quality of life; therefore, it is necessary to promote health via the fundamental tool of nutritional literacy. In this context, this study aims to evaluate nutritional literacy in Peru through the S-NutLit tool, which is composed of two dimensions. The scale, composed of 11 items, was applied to 396 Peruvian adults. According to the evaluation of the indicators, an acceptable reliability was found, as was a model fit with excellent estimation according to its indicators (CMIN/DIF 2.559; IFC 0.965; SRMR 0.043; RMSEA 0.063; and PClose 0.070). In this way, we seek to reinforce health promotion activities through a nutritional literacy diagnosis, which, due to its characteristics, can be self-administered and used by health entities and other entities in general that are interested in knowing the eating practices of an individual, which undoubtedly leads to good health.

## 1. Introduction

Nutritional literacy is defined as the degree to which an individual acquires, processes, and understands basic nutrition information and services needed to make adequate, informed, and aware nutrition decisions [1]. It is an approach that addresses complex public health problems by focusing on an individual’s abilities and skills to choose healthy foods [2]. It is also considered as one of the factors that helps to maintain good practices regarding the consumption of adequate food; that is, it allows people to have information regarding the risks generated due to poor nutrition. This term is conceptualized as the union of educational strategies, which is accompanied by environmental support, whose purpose is to voluntarily promote a change in food alternatives and other changes that are linked to food and nutrition that lead to the well-being and an improvement of health in the population [3]. It is also referred to as those cognitive and social skills that motivate an individual to know, process, and practice information regarding how to improve health through practices that contribute to the maintenance of good health [4]. Additionally, it is also considered as a concept that is closely related to behavior concerning personal lifestyle practices and the way in which a person may interact with the health system [5].

Contrary to this, nutritional illiteracy leads to poor daily dietary choices, which increase the development of diseases [6]. Under this approach, individuals have the option to make more informed decisions regarding their diet, since the root cause of many diseases is precisely poor nutrition [7]. According to Pi et al. [8], the most common gastrointestinal diseases rank third in incidence and are the fourth most deadly among malignant tumors worldwide; this statistic highlights how important nutritional literacy can be in health care and disease prevention.

Some of these diseases originate from lifestyle, genetics, socioeconomic conditions, and other factors. According to recent research, it is stated that a poor diet can affect an individual’s health [9]. Although this statement is known by the general population, few people are truly aware of the modifications that need to be made to their unhealthy lifestyle, which is why they fall back into some diseases. Thus, health systems have become an important entity to mitigate, to some extent, the appearance of some diseases [10]. When a person gets sick, he/she seeks a medical professional for a prescription, allowing for the possibility that the attending physician can directly influence the patient regarding nutritional practices. However, this opportunity has been in vain, since it has been identified that these professionals still lack the necessary skills to accurately recommend any change in the dietary routines of the patient and that within the curriculum of physicians, a subject oriented to nutritional literacy has not yet been established [11].

As a result of what was mentioned in the previous paragraph and taking into consideration that food has the potential to generate the welfare of the population as a whole, a new alternative has emerged within medical environments: healthy lifestyle medicine. Promoting healthy actions and/or certain changes to the diet allows control of the onset of some diseases, thus improving the overall welfare of the population [12]. In addition to this, other studies recommend that literacy regarding nutrition and a healthy lifestyle should begin in early childhood, as educational environments are ideal spaces to instill healthy eating habits and promote physical activity. However, this practice could not guarantee a good feeding practice, as it requires the intervention of professionals who can instruct parents and the intervention of an educational institution to act as supporting agents in order to encourage parents to adopt good feeding practices through nutritional literacy [13,14,15].

The action of obtaining and ensuring certain adequate nutritional knowledge is a nutritional literacy activity [16]. Thus, researchers have developed measures to assess nutritional literacy in order to provide an important contribution to diagnose the level of this variable of study, with the purpose of promoting the creation of effective programs to ensure the acquisition of adequate nutritional knowledge by the population. In this regard, De la Fuente-Anuncibay et al. [17] evaluated this variable through knowledge of the food pyramid, the food information channel, and perception of food culture. On the other hand, Buczak [18] developed a “concise scale of food attitudes” in order to diagnose attitudes towards nutrition in adults through an instrumental study. For this, he applied 17 items which were made up of the following dimensions: anorectic attitude (4 items), experimental attitude (2 items), hedonistic attitude (3 items), orthorectic attitude (3 items), religious attitude (3 items), and vegetarian attitude (3 items).

Although there are studies on nutritional literacy, there are researchers who recognize nutritional literacy as nutrition education. For example, a Polish version of the Short Food Literacy Questionnaire (SFLQ) consists of three dimensions: “access to information”, “knowledge” and “evaluation of information”, consisting of four, three, and four items, respectively. Meanwhile, unidimensional metrics that measure the same variable have also been identified, and each metric depends on the context in which it is applied. Evidence of this is the study of variable food literacy in the Turkish version of SFLQ, which is constituted with a unidimensional structure made up of 12 items [19]. It is similar to the Swiss version, which also consists of 12 items, and maintains the same behavior data [2].

Studies indicate that in nutritional literacy, the information skills and expert skills are essential to ensure that people can access, understand, and use accurate and relevant nutritional information. Vrinten et al. [20] states that information skills refer to the ability to find, evaluate, and use information effectively. In the context of nutritional literacy, these skills are essential for people to make informed decisions about their dietary choices and overall health. Another important factor is expert skills, which go beyond basic information skills. These include a deeper understanding of scientific principles and the practical application of nutritional knowledge. People with these skills can contribute to this area, provide guidance, and make informed decisions [20,21,22]. In general, information skills provide a foundation for basic nutrition skills, while expert skills elevate a person to a level where they can apply advanced knowledge and contribute significantly to the field of nutrition. Both skills are essential to develop a population that understands nutrition and can make healthy decisions [20,23].

In addition, research linked to food attitudes indicates that it is important for consumers to find and understand the nutritional information of the products they purchase. Thus, the food industry, in its attempt to contribute to nutritional literacy, makes public information available regarding nutritional values, ingredients, and recommendations on product consumption [24]. However, it seems that efforts to acquire a balanced diet through good nutritional literacy are not sufficient, as reflected in the high rates of patients who come to a health center for care and do not receive the necessary preventive information [25]. Hence, it is important to take into account that contributing to a healthy society is everyone’s task, since, according to previous research, nutritional literacy symbolizes a significant change in people’s knowledge and attitudes [26]. Thus, it is important that in addition to health professionals, every individual can maintain a certain level of nutritional literacy in order to support disease prevention or combat diseases.

Now, considering that the first step for an individual to engage with a sustainable food system is to acquire skills that allow him/her to have a healthy relationship with food [27], it is necessary to measure these skills through a nutritional literacy questionnaire. For this reason, this study was designed for use in the Peruvian population, which is currently facing very significant challenges regarding good eating practices. Although the Peruvian state has established certain policies and has been promoting nutritional literacy, there are still some gaps to be covered, for example, the lack of an accurate diagnosis that highlights the nutritional literacy of the population. Thus, this study aims to translate and validate the properties of the S-NutLit nutritional literacy questionnaire to fill said gap, which is considered an appropriate intervention. 

### Nutritional Literacy Scales

Nutritional literacy measurement scales are important because they are crucial tools for assessing and understanding the level of knowledge, discernment, and understanding that people have about nutritional information and food-related decisions [22,28]. Its necessity is based on several reasons, such as assessing nutritional understanding, detecting gaps in knowledge, developing effective educational programs, improving decision making, preventing diet-related diseases, and empowering individuals [29,30]. In such a matter, nutritional literacy measurement scales are essential to understand and address personal, occupational, community, and governmental challenges, thus enabling the development of more effective interventions and programs to promote health and wellness.

On this topic, according to the Nutbeam model [20,21,22,31], functional literacy is understood as the ability to access, understand, and use food and nutrition information, including knowledge of a variety of food and nutrition topics, and the practical skills needed to obtain, select, prepare, and eat healthy foods. Interactive literacy is the ability to exchange information, share information, discuss food and nutrition information with others, and engage in collaborative activities. And critical literacy is known as the ability to critically evaluate food and nutrition information, recognize the impact of diet and food choices on society, understand food as an integral part of complex production and distribution processes, and recognize the impact of various social conditions and behaviors.

Nutritional literacy has been associated with beneficial health outcomes for individuals [20]. In response to this demand, several scales have been designed to measure this construct. One of these measurement scales that has been most widely adapted to various countries is the Nutrition Literacy Scale (NLS), originally designed by Diamond [32] in adult patients in the U.S.A., and later validated and adapted by Patel et al. [33] (African Americans), Zanella et al. [34] (Brazil), Guttersrud et al. [35] (Norway), Michou et al. [28] (Greece), and Coffman and La-Rocque [36] (various Latin countries). The Nutrition Literacy Assessment Instrument (NLit) of Gibbs and Chapman-Novakofski [29,37] was originally developed in adult patients from the U.S.A. and was later adapted to the Latino population of mostly Mexican descent residing in the U.S.A. [38], Brazilians [39], Italians [40], and Chinese diabetic patients [41]. The S-NutLit scale is a short nutritional literacy tool [20] composed of 11 items and divided into two dimensions: informational skills (α = 0.83 and composed of 8 items) and expert skills (α = 0.79 and composed of 3 items). The following table (Table 1) describes various measurement scales for nutritional literacy.

After a diligent review of the aforementioned background, there has been significant interest in developing scales to measure the nutritional literacy construct. Previous research has designed scales to measure this construct in countries such as the U.S.A., Iran, Korea, Thailand, Netherlands, and China. These scales have been applied to various sectors and study populations, such as adult patients [29,32], patients with renal diseases [31], pregnant women [48], the elderly [49], children and adolescents in basic education [22,42,44,46,47], undergraduates [21,43], and young adults [20,45]. The ten countries that have emphasized this construct the most in their studies have been the United States, China, Iran, Turkey, Australia, Greece, Norway, Taiwan, Qatar, Italy, and Brazil. Meanwhile, in Peru there is no Spanish version in the scientific literature with evidence of the validity and reliability of a nutritional literacy scale. To fill this knowledge gap, it is necessary to conduct a study to adapt the S-NutLit scale (of Dutch origin) for economically active Peruvian adults. In this sense, the validation of the 11-item S-NutLit scale was considered appropriate.

## 2. Materials and Methods

This research aimed to evaluate the validity and reliability of the instrument that measures nutritional literacy, which was initially designed to measure this variable in young adults and was proposed by Vrinten et al. [20]. The S-NutLit scale is composed of 11 items, which are evaluated with a Likert scale of 1–5, where 1 represents totally disagree and 5 represents totally agree.

### 2.1. Validation of the S-NutLit Instrument

The S-NutLit scale was developed and validated by Vrinten et al. [20], who, after reviewing the literature, submitted the instrument to expert judgment evaluation to validate the content, interview the study population, reduce the number of items, and validate the instrument through an exploratory factor analysis and reliability tests. This resulted in a short instrument of nutritional literacy for young adults, consisting of 11 items and divided into two dimensions: information skills (α = 0.83 and consisting of 8 items) and expert skills (α = 0.79 and consisting of 3 items). The study participants consisted of 300 young adults with a mean age of 21.6; moreover, 28% of the population had some active link with the field related to health or nutrition. The statistical treatment indicated the elimination of one item because it did not comply with the determined limits, thus the scale was made up of 11 items whose subscales explained 43% of the total variance.

In order for this same instrument to be applicable to the Peruvian population, a back-translation process was carried out; that is, the selected questionnaire was translated by a bilingual professional with translation experience from English to Spanish and Spanish to English (being that the original was developed and published in English). This method avoided any discrepancy or loss of meaning of the initial instrument [50]. Subsequently, the focus group participants were selected and were invited to join a Zoom room where the translated instrument was presented in order to receive their opinions regarding each item. The session lasted 60 min, during which the participants gave semantic validity to the instrument, thus providing evidence that each of the items was understandable and suitable for application in the Peruvian population. It should be noted that the focus group was made up of representative participants of the sample, thus involving 6 Peruvian adults including 1 housewife, 2 university students, 1 Peruvian professional, and 2 independent workers. Furthermore, these 6 participants were established under the criteria supported by Krueger and Crasey [51], who suggest that this size is adequate to obtain different opinions and experiences from the population, thus ensuring the clarity and understanding of the instrument to be applied [52]. In the session, the original instrument had three changes: in NL3, the original instrument says “*When searching for nutritional information on the Internet, I can distinguish between reliable and less reliable websites*”, which was changed to “*When I search for nutritional information on the Internet, I can distinguish between reliable and less reliable websites*”; in NL5, the original instrument says “*I have the necessary skills to apply nutrition information when cooking*”, which was changed to “*I have the necessary skills to apply nutritional information when cooking*”; and in NL10, the original instrument says “*I discuss nutritional information with an expert*”, which was changed to “*I discuss nutritional information with experts*”. As shown, the modifications were minimal because the members of the focus group managed to clearly understand each of the items.

### 2.2. Data Collection

The questionnaire was elaborated in Google forms (Google Inc., Menlo Park, CA, USA), thus generating a link that was shared through social networks such as WhatsApp (WhatsApp Inc., Menlo Park, CA, USA) and Telegram (Telegram Messenger Inc.Tortola, British Virgin Islands). In the survey, informed consent was included in the first part; that is, each of the participants, prior to answering the survey, gave their consent regarding knowledge of the study, its use, and its purpose. Therefore, the study population participated freely, voluntarily, and without a time limit. Additionally, this study was previously evaluated and approved by the ethics committee of the Peruana Union University (Universidad Peruana Union), thus guaranteeing the scientific quality and well-being of the participants. The inclusion criteria were participants who declared that they were consumers of healthy foods, of a legal age (18 years and older), and who had higher education (at least a minimum level of education). A total of 396 valid data were collected.

### 2.3. Statistical Analysis

The study began with a descriptive analysis of the items in a sample of 400 participants, evaluating the mean, standard deviation, asymmetry, and kurtosis of each item. For skewness and kurtosis, acceptable ranges were those close to zero, ideally between −1.5 and +1.5, indicating an approximately normal distribution [53].

As a second step, an exploratory factor analysis (EFA) was performed in SPSS-V25, implementing the maximum likelihood method and a Promax rotation. Prior to EFA, the suitability of the data for factor analysis was assessed by applying the Kaiser–Meyer–Olkin (KMO) measure and Bartlett’s test of sphericity. The KMO value was found to be a minimum acceptable value of 0.70, indicating the adequacy of the factor analysis. Bartlett’s test of sphericity was expected to have a *p*-value of less than 0.05 to significantly reject the null hypothesis that the variables are not correlated in the space of variables.

Following these steps, a stage of data analysis that included a confirmatory factor analysis (CFA) was performed with Analysis of Moment Structures (AMOS-V24) software as part of covariance structural equation modeling (CB-SEM). It was anticipated that the factor loadings would be significant to confirm the convergent validity of the constructs, with an average variance extracted (AVE) exceeding the threshold of 0.50, as recommended by Fornell and Larcker [54]. For reliability, both Cronbach’s alpha and composite reliability were expected to exceed a value of 0.70, indicative of acceptable reliability [55,56]. For the discriminant validity test, the heterotrait–monotrait ratio (HTMT) criterion was applied, where values below 0.90 are expected [57].

Finally, the model was tested. The model fit indices, which include the Comparative Fit Index (CFI), the Root Mean Squared Error of Approximate Error (RMSEA), the Standard Model Fit Index (SRMR), the likelihood ratio of closeness (PClose), and the chi-square ratio over degrees of freedom (CMIN/DF), were expected to meet the recommended standards. These standards were CFI > 0.95, RMSEA < 0.06, SRMR < 0.08, PClose > 0.05, and CMIN/DF < 3, which would indicate a good model fit [58].

## 3. Results

Regarding the descriptive statistics, 396 people, aged 18–56 years, of whom 154 were men and 242 women, participated in the study. Regarding marital status, 27 were married, 4 were cohabiting, 5 were divorced, 359 were single, and 1 was widowed. Considering that the minimum living wage (RMV) in Peru is Peruvian Soles-PEN 1025.00, the study population was asked how much their monthly income was in terms of RMV. The result was that 3.3% of the families earned between 11 and 20 minimum wages, which is between level A and B according to the Peruvian socioeconomic level; 22% earned between 3 and 4 minimum wages, thus this population was located at socioeconomic levels C and D; 11.9% earned between 5 and 10 wages and represented those who belong to socioeconomic level B; the highest percentage (60.9%) earned up to 2 minimum wages and represented those who belong to economic level E; and finally, 2% earned more than 20 minimum wages, thus according to their income they are located in socioeconomic level A.

Table 2 shows the descriptive statistics of the S-NutLit scale items (mean, standard deviation, skewness, and kurtosis). It was observed that all skewness and kurtosis values were less than +/−1.5 [59], which meets the multivariate normality assumption.

Table 3 shows the exploratory factor analysis (EFA) of the items, where it can be clearly observed that the items are distributed in two factors, thus confirming the original distribution of the S-NutLit scale. The total variance explained in the model is 51.6%, which is greater than 50%, with the Factor 1 information skill representing 46.3% and the second factor knowledge skill representing 5.2%. With these results and to continue with the validation process, the confirmatory factor analysis (CFA) was performed. 

The Kaiser–Meyer–Olkin test was greater than 0.70 (KMO = 0.927), which is high, and the Bartlett test was highly significant (Sig = 0.000), so the exploratory factor analysis (EFA) could be performed. The maximum likelihood method was chosen and found that the 11 items were distributed in two factors, thus confirming the original distribution of the S-NutLit scale. The total variance explained in the model was 51.6%, which is greater than 50%, with the Factor 1 information skill representing 46.3% and the second factor knowledge skill representing 5.3%. With these results and to continue with the validation process, the confirmatory factor analysis (CFA) was performed (See Table 3).

From the results of the exploratory factor analysis, this study found the distribution of two factors: for information skills, the items NL1, NL2, NL3, NL4, NL5, NL6, and NL8; and for knowledge skills, the items NL7, NL9, NL10, and NL11. This was compared to the original instrument, whose distribution for the first factor was NL1, NL2, NL3, NL4, NL5, NL6, NL7, and NL8, while the second factor was NL9, NL10, and NL11. This means that the divergence suggests a new way of distributing the structure of the instrument, so there is a high possibility that respondents have identified a different view of nutritional literacy compared to the original instrument.

On the other hand, the evaluation of the reliability of the items had a Cronbach’s alpha of 0.863 for the first factor and 0.831 for the second factor. Likewise, the CR had values of 0.864 and 0.835 for the first and second factors, respectively; these values guarantee the internal consistency of the items for each dimension of the scale. In addition, the AVE (Average Variance Extracted) that evaluates the convergent validity had a value of 0.651, which was an adequate value. The heterotrait–monotrait relationship analysis (HTMT) was also performed to evaluate the discriminant validity between both factors of the scale [57]; to comply with this indicator, the coefficients must be below the strict point (0.850). In the case of the present study, this criterion was met by having a value of 0.826 (see Table 4).

Concerning the model fit, it is shown that all the indicators had an excellent estimation, and only the RMSEA indicator was at an acceptable level of 0.063 [58] (see Table 5, Figure 1).

Due to previous evidence by Vrinten et al. [20] and the exploratory factor analysis performed (Table 3), a two-factor internal structure was considered for the confirmatory factor analysis (model 1). However, to verify the fit of the model, a unidimensional analysis of the 11 factors was performed to explain a single factor (model 2), and it was found that the model with a single factor did not achieve a good fit, so the initial model (model 1) with two factors was considered the most appropriate for the S-NutLit scale. In addition, the correlation between its two internal factors was 0.826, which is in the strictest range of discriminant validity with a value of 0.850 [57] (See Table 5). It should be noted that in the analysis of the original instrument, a correlation of R = 0.2 was found. While this study obtained an indicator of 0.81, this disparity in the correlations could be explained by the differences in the sample according to the demographic, cultural, and economic characteristics in which the study was applied, and may even be due to the conceptualization of the terms since there are studies which have established that eating habits measure eating behaviors [60].

## 4. Discussion

The objective of this study was based on translating and validating the nutritional literacy scale (S-NutLit) in the Peruvian context in order to provide the population and the scientific community with a short scale that promotes health in the Peruvian population and that can also be replicated in other contexts. For this purpose, a back-translation and semantic validation procedure were carried out, resulting in the design of an applicable, understandable instrument in a reduced version which will allow the population to self-administer a diagnosis regarding the conditions of nutritional literacy. The diagnosis made according to this questionnaire could symbolize a determinant for the population to become aware of the importance of knowing about nutrition issues that lead them to adopt certain practices that contribute to a healthy diet that allows for the preservation of good health. 

This study has shown that there is a difference in the distribution of the items when comparing item NL7 in the original instrument (“I know the basic rules of the Flemish Food Triangle”) with the Spanish version (“I know the basic rules of the Food Triangle”). In the case of the instrument validated in the Peruvian context, item NL7 is part of the expert skills subscale, when it was originally part of the information skills subscale. An explanation for this fact is that although the Food Triangle had a significant change in September 2017, the modification of eating behavior in the population was not a guarantee, since an additional intervention was required to understand the adoption of new dietary guidelines [61]. In addition to this, studies that explore or explain the gaps between the Food Triangle and the Flemish Food Triangle are still scarce, which means that few people have learned what the Flemish Food Triangle consists of. In this context, the experts agreed to remove the term “Flemish” and leave the statement as the traditionally known “Food Triangle”. Another concept that could explain this fact is that the term healthy plate has currently gained greater ground in terms of studies that act as new perspectives for nutritional intervention [62].

On the other hand, the scientific literature refers to nutritional literacy as the ability of an individual to know information regarding nutrition, which also implies the ability to make informed decisions about how to eat [63,64]; thus, it is considered that nutritional literacy is useful to implement good habits, which are consolidated during daily life [65,66]. In addition, the fact that individuals maintain a good nutritional literacy symbolizes that they can avoid excessive consumption of some food of their preference, or otherwise decrease the quantity and quality of food in their daily diet [49,67]. Given this importance, there is research that focuses on measuring the ability of a consumer to understand the information of a product; thus, the nutritional literacy scale constitutes two dimensions. This multidimensional distribution is also supported by Nooriani et al. [68], which sustains that the nutritional literacy scale presents the following dimensions: perceived susceptibility, perceived severity, perceived benefits, perceived barriers, and self-efficacy. Meanwhile, [69] posits that the same nutritional literacy scale is made up of nutritional functions, food sources, and food consumption habits. Additionally, the identification of unidimensional metrics is highlighted as they correspond to a single factor [2,19,70]; in each of them, it is also possible to determine the purchasing behavior of the consumers [71]. All of the above is evidence of the concern of the scientific community to make a measuring tool that identifies levels of nutritional literacy available to the population, as it is assumed that a low level of this could lead to unhealthy dietary practices and nutritional status [33].

According to the results, the analyzed scale had two dimensions and was made up of 11 items, and when the intention was to measure the perspectives of a population, research established the need to apply a short questionnaire that was easy to administer and that could allow for greater participation by the study population, thus obtaining a larger representative sample that allowed for solid, generalizable conclusions with less risk of research bias. In this sense, the number of items does not compromise the validity of the construct [72,73]. This represents a significant contribution, since the administration of the questionnaire requires only a few minutes and allows the nutritional literacy of the population to be diagnosed immediately. This symbolizes an economic saving for institutions wishing to measure the conditions of their workers, and more broadly is useful for any organization that is interested in measuring this variable of study. Considering these results, regardless of the economic benefit (savings), the institutions can intervene effectively through policies based on current information that lead to the maintenance of good nutritional practices. 

Finally, it was confirmed that S-NutLit is a reduced scale and applicable in the Peruvian context, and its reliability indicators are within the determined ranges and divided into two factors: informational skills (α = 0.83) and expert skills (α = 0.79), which makes the instrument a valid metric. In this regard, theoretical foundations have been identified that support that informational and expert skills are part of nutritional literacy, described as the ability of an individual to obtain, understand, and comprehend nutritional information, which leads to improvement of an individual’s behavior towards informed and healthy eating [1,74]. Thus, the division of the factors allows a better view of the multidimensional nature of the construct, which shows that beyond the population’s knowledge of nutritional information, it also extends to knowing how to interpret and apply it in daily life. 

According to other studies, there are scales that evaluate this construct; for example, Ahn et al. [45] establishes that the metric for nutritional literacy in young adults is the Nutrition Literacy Questionnaire (NLQ), which is made up of 30 items, divided into six dimensions, whose reliability is α = 0.87. In this case, the author reports a questionnaire where nutritional literacy is measured through nutritional labeling and nutritional management. On the other hand, the metric designed and applied by Liu et al. [47] in China focused on measuring the perspectives of school-age children and adolescents; this metric is called the Food and Nutrition Literacy Questionnaire (FNLQ-SC) and is made up of 19 items and five dimensions and whose alpha values correspond to 0.698. This measure is considered by the author as a modifiable factor essential for health promotion. In both cases, the measures are related to the evaluation of the skills, knowledge, and attitudes of consumers regarding their diet. 

Additionally, the Nutritional Literacy Scale, made up of four dimensions and 28 items, presents an alpha of 0.830 and was applied to Chinese patients with renal diseases. Similarly, Mo et al. [21] support the Short-Form NL scale (NL-SF12), whose objective is to measure nutritional literacy in Chinese university students. This scale has six dimensions and 12 items with an acceptable reliability of 0.89. In all cases, researching, creating and applying a scale to measure nutritional literacy is a key element that can be applied to any context as long as the scale meets the minimum criteria of applicability. Thus, this principle demonstrates the diversification of scales that address the subject with the common objective of understanding and addressing certain practices related to food, thus emphasizing the need to continue discovering and updating knowledge regarding the practical skills of food and nutrition.

### 4.1. Theoretical Implications

Focusing on nutritional literacy has become a strategy to improve and develop people’s physical well-being and make a country’s health more stable. This study contributes to the literature by carefully developing an overview of the construct supported by recent research. In addition, it also provides a sufficiently solid scientific basis to offer suggestions for the implementation of strategies focused on nutritional literacy. The results demonstrate the scope and expansion of nutritional literacy and its impact on a population. In this sense, deepening its theoretical approach not only strengthens the quality of the research, but also improves the applicability and relevance of the results in practical and clinical contexts. Furthermore, it contributes to the advancement of knowledge in the field and provides a strong basis for future studies. 

### 4.2. Practical Implications

Researchers and health professionals have a brief metric that can be easily incorporated in the exercise of their profession and necessary applications. As a product of this study, it has been suggested that the nutritional literacy scale (S-NutLit) in the Peruvian context promotes the improvement of quality of life, a healthy diet, and the development of good eating habits. In addition, the results of this study suggest that improving nutritional literacy could prevent health risks and complications (diabetes, malnutrition, insufficient intake, and poor quality of life), leading to a modification of one’s lifestyle to prevent the onset and progression of these health disorders [23,74,75].

Since this research confirms the validity and reliability of the scale, it is important that future studies take into account the effect of nutritional literacy on chronic diseases associated with nutrition and on optimal decision making. Likewise, it is possible for the presented scale to be self-completed in a short time, which allows it to be very well-used in future studies. In this sense, the proper use of this tool could be a valuable contribution to the scientific community and to those in charge of the Ministry of Health and other public and private organizations to continue scrutinizing nutritional literacy and improving public health in the country. In addition, it could facilitate the identification of people with inadequate nutrition knowledge or those at risk of having it and design effective strategies with the purpose of addressing this need. When reflecting on policy decisions focused on the nutrition of citizens, it is suggested that these should provide the community with clear and useful information about food.

Consequently, this study deepens the knowledge on nutritional literacy, which could allow the top management of any organization (public or private) to consider renewing new ways and strategies to improve the health of more citizens, even more so in the Peruvian context, which is where the development of the S-NutLit scale acquires high relevance because in this country, food is considered as a cultural construct based on tradition, custom, and belief, which means that in addition to nutritional considerations, cultural factors are implicit in food consumption decisions. Therefore, the S-NutLit scale in the Peruvian context provides a better understanding of nutritional education and traditional food practices.

## 5. Conclusions and Limitations

This research demonstrates the distribution of the items that make up the S-NutLit metric to measure nutritional literacy, which has a distribution divided into two factors which have high levels of reliability, with a Cronbach’s alpha of 0.906 and a CR of 0.910. As these indicators are higher than 0.70, they are qualified as reliable and valid. In addition to these high-value indicators being qualified as reliable and valid, their AVE of 0.518 means that more than 50% of the variance in the items corresponding to the questionnaire is related to the construct, supporting a high validity of the instrument. 

Regarding the limitations of the research, although 396 surveys were collected, the results cannot be generalized because when applied in different contexts, it is necessary to carry out the necessary process to demonstrate the validity of the survey. The distribution of data differs from the original instrument, so it is established that the S-NutLit scale could not be universal.

## 6. Future Research

The distribution of the items differs from the original study, so that the variability becomes an opportunity for future studies to evaluate how the distribution of items could impact the interpretation of the results. In addition, this study also proposes future research to apply the scale to other Latin American contexts in order to identify differences in item reliability and whether the distribution is maintained. 

## Figures and Tables

**Figure 1 ijerph-21-00707-f001:**
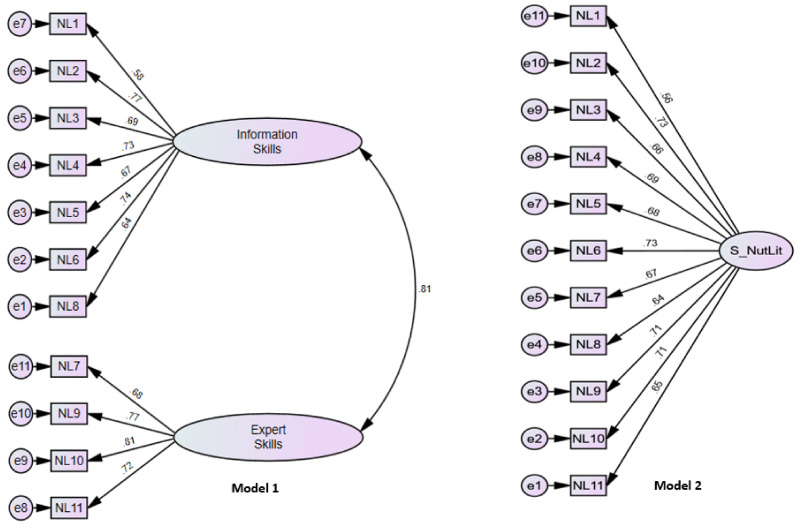
Measurement model to assess nutritional literacy in the Peruvian context. E = measurement errors.

**Table 1 ijerph-21-00707-t001:** Measuring scales for nutritional literacy.

Scale name	Author(s)	Country	No. of Items	Dimensions	Population	Alpha
Nutrition Literacy Scale (NLS)	Diamond [32]	U.S.A.	28	(1) Nutritional knowledge, (2) healthy nutrition, (3) calorie use, (4) organic food, (5) saturation fats, and (6) portion size.	Adult patients	0.84
NLit	Gibbs and Chapman-Novakofski [29]	U.S.A.	35	(1) Nutrition and health, (2) energy sources in food, (3) food label and numeracy, (4) household food measurement, (5) food groups, and (6) consumer skills.	Adult patients	NS
Student Nutrition Literacy Survey (SNLS)	Hawkins et al. [42]	U.S.A.	18	(1) Nutrition knowledge, and (2) attitudes, beliefs, intent (ABI).	Primary school students	NS
Young Adult Nutrition Literacy Tool (YA-NLT)	McNamara et al. [43]	U.S.A.	42	(1) Functional, (2) interactive, and (3) critical.	University students	NS
Food and Nutrition Literacy (FNLIT)	Doustmohammadian et al. [44]	Iran	42	(1) Understanding, (2) knowledge, (3) functional skills, (4) food choice skills, (5) interactive skills, and (6) critical skills.	Primary school children	Between 0.48 and 0.80
Nutrition Literacy Questionnaire (NLQ)	Ahn et al. [45]	Korea	30	(1) Dietary guidelines, (2) nutrition and health, (3) nutrients, (4) five food groups, (5) nutrition labeling, and (6) nutrition management for disease prevention.	Young adults	0.87
Thai Nutritional Literacy Assessment Tool (Thai-NLAT)	Deesamer et al. [46]	Thailand	61	(1) Micronutrients and health, (2) nutrition and energy balance, (3) decision making on nutrition information, (4) food processing, and (5) food safety.	Teenagers	NS
Short Nutrition Literacy (S-NutLit)	Vrinten et al. [20]	Netherlands	11	(1) Information skills and (2) expert skills.	Young adults	0.79 y 0.83
Nutritional Literacy Scale (NLS)	Li et al. [31]	China	28	(1) Nutrition knowledge level, (2) cognitive and attitude, (3) behavior practice ability, and (4) information acquisition ability.	Patients with kidney disease	0.83
Food and Nutrition Literacy Questionnaire for Chinese School-age Children (FNLQ-SC)	Liu et al. [47]	China	19	(1) Knowledge and understanding, (2) access to and planning for food, (3) selecting food, (4) preparing food, and (5) eating.	School-age children and adolescents	0.698
Nutrition Literacy Assessment Instrument for Chinese Pregnant Women (NLAI-P)	Zhou et al. [48]	China	38	(1) Knowledge, (2) behavior, and (3) skill.	Pregnant women	0.82
Chongqing Middle School Student Nutrition Literacy Scale (CM-NLS)	Wang et al. [22]	China	52	(1) Obtain, (2) understand, (3) apply, (4) interact, (5) medial literacy, and (6) critical skills.	High school students	0.849
Short-Form NL Scale (NL-SF12)	Mo et al. [21]	China	12	(1) Knowledge, (2) understanding, (3) obtaining skills, (4) applying skills, (5) interactive skills, and (6) critical skills.	University students	0.89
Nutrition Literacy Questionnaire for the Chinese Elderly (NLQ-E)	Aihemaitijiang et al. [49]	China	25	(1) Knowledge, (2) understanding, (3) dietary behavior, (4) healthy lifestyle, (5) cognitive skills, and (6) operational skills.	Elders	0.678

NS: Not specified.

**Table 2 ijerph-21-00707-t002:** Descriptive statistics of S-NutLit.

	Median	Standard Deviation	Skewness	Kurtosis
NL1	3.61	0.89	−0.407	0.120
NL2	3.66	0.94	−0.522	−0.045
NL3	3.59	1.01	−0.563	−0.016
NL4	3.56	0.93	−0.527	0.268
NL5	3.51	0.96	−0.389	−0.088
NL6	3.50	0.96	−0.445	−0.075
NL7	3.51	1.05	−0.561	−0.143
NL8	3.60	0.94	−0.388	−0.024
NL9	3.52	0.96	−0.393	−0.046
NL10	3.17	1.08	−0.236	−0.456
NL11	3.07	1.04	−0.164	−0.395

**Table 3 ijerph-21-00707-t003:** Exploratory Factor Analysis (EFA) Pattern Matrix.

Items		Factor	
		1	2
NL2	Si tengo preguntas sobre nutrición saludable, sé dónde encontrar información al respecto. If I have questions about healthy nutrition, I know where to find information about it.	0.772	
NL3	Cuando busco información nutricional en Internet, puedo distinguir entre sitios web confiables y menos confiables. When I search for nutritional information on the Internet, I can distinguish between reliable and less reliable websites.	0.762	
NL4	Si tengo dudas sobre nutrición sostenible, sé dónde encontrar información al respecto. Ejemplos de nutrición sostenible son las verduras orgánicas, los huevos de gallina criados en libertad, el café de comercio justo, etc. If I have questions about sustainable nutrition, I know where to find information about it. Examples of sustainable nutrition are organic vegetables, free-range chicken eggs, fair trade coffee, etc.	0.750	
NL6	Los anuncios a menudo establecen una conexión entre la nutrición y la salud. Me resulta fácil juzgar si estos enlaces son ciertos o no. Advertisements often make a connection between nutrition and health. It is easy for me to judge whether these links are true or not.	0.608	
NL1	Puedo evaluar si la información sobre nutrición en los medios es confiable. I can assess whether nutrition information in the media is reliable.	0.559	
NL8	Puedo evaluar si la información sobre nutrición está escrita con la intención de ganar dinero, por ejemplo, por personas que quieren vender un producto. I can assess whether nutrition information is written with the intention of making money, for example, by people who want to sell a product.	0.498	
NL5	Tengo las habilidades necesarias para aplicar la información nutricional al cocinar. I have the necessary skills to apply nutritional information when cooking.	0.378	
NL10	Hablo de la información nutricional con un experto. I discuss nutritional information with an expert.		0.844
NL11	Mi dieta está basada en los últimos conocimientos científicos. My diet is based on the latest scientific knowledge.		0.754
NL9	Sigo los consejos de nutrición de los expertos. I follow the nutrition advice of the experts.		0.740
NL7	Conozco las reglas básicas del Triángulo Alimentario. I know the basic rules of the Food Triangle.		0.530

Note: Extraction method: maximum probability. Rotation method: Promax with Kaiser normalization.

**Table 4 ijerph-21-00707-t004:** Validation of the measurement model and the convergent and discriminant validities.

Predictor	Outcome	Std Beta	Alpha	CR	AVE	HTMT
Information Skills	NL8	0.644 ***	0.863	0.864	0.651	0.826
	NL6	0.744 ***				
	NL5	0.669 ***				
	NL4	0.732 ***				
	NL3	0.691 ***				
	NL2	0.765 ***				
	NL1	0.579 ***				
Expert Skills	NL11	0.723 ***	0.831	0.835	0.651	
	NL10	0.807 ***				
	NL9	0.773 ***				
	NL7	0.684 ***				

Note: *** *p* < 0.001 (significance level).

**Table 5 ijerph-21-00707-t005:** Model fit.

Measure	Threshold	Model 1		Model 2	
		Estimate	Interpretation	Estimate	Interpretation
CMIN	--	110.045	--	218.996	
DF	--	43.000	--	44.000	
CMIN/DF	Between 1 and 3	2.559	Excellent	4.977	Acceptable
CFI	>0.95	0.965	Excellent	0.909	Acceptable
SRMR	<0.08	0.043	Excellent	0.058	Excellent
RMSEA	<0.06	0.063	Acceptable	0.100	Terrible
PClose	>0.05	0.070	Excellent	0.000	Not Estimated

## Data Availability

Data are available upon request from the authors.

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
