# Peer review of "Adaptation and Validation of the S-NutLit Scale to Assess Nutritional Literacy in the Peruvian Population"

_ijerph, 2024, doi:10.3390/ijerph21060707_

Round 1

Reviewer 1 Report (Previous Reviewer 1)

Comments and Suggestions for Authors

Lines 165:  Change NLit of Gibbs & Chapman-Novakofski to the instrument name which is "The Nutrition Literacy Assessment Instrument (NLit)"

Line 165: remove it after year 2012,2013

Line 166-167: Correct Spanish-speaking population to Latino, mostly Mexican heritage, Spanish-speaking residing in the USA

Line 168: It sounds out of context, I cannot understand what the phrase in this sense is referring to

Table 1: you need to add all the NLit instruments cited on lines 165 to 168

Lines 184-187: You need to add Italy to the list of countries

Lines 281-284:  Please interpret the income ranges. How many/percentage of your sample were low and high-income? 

Lines 297 - 304: Keep to single digit after dot. For example, 51.6% instead of 51.620%

Comments on the Quality of English Language

This manuscript needs a review of academic English writing to improve cohesiveness and comprehension.

Author Response

Dear Reviewer,

Thank you very much for your informed comments, which helped us so much in improving the manuscript. We appreciated the time you spent doing this and tried our best to address all your comments. We hope that this revised version of the paper reaches the expected standard, worthy of publication in this journal.

A detailed list of answers to your comments and suggestions is reported below.

Many thanks for your time.

Best regards,

Reviewer 2 Report (Previous Reviewer 2)

Comments and Suggestions for Authors

In summary, this study provides valuable and enlightening insights into the psychometric characteristics of the Peruvian adaptation of the nutritional literacy tool, known as the S-NutLit Scale. I am delighted with the incorporation of my previous recommendations pertaining to the manuscript.

The authors have provided a clear and concise overview of nutritional literacy and its measures. The aims of the study are well-formulated, and the study design and data analysis appear to be appropriate. This paper makes an important contribution to the literature by showing that the questionnaire is a useful measurement tool in other cultures.

As a critique, I can reiterate certain remarks made previously:

When considering CFA, it is beneficial to include alternative models, since the fit indices can be effectively understood in relation to this, such as a one-dimensional structure or a 2-dimensional structure in a second-order or bifactor arrangement. Merely possessing satisfactory fit indices does not substantiate the notion that it does not enhance the quality of the model. Given the strong correlation (r = 0.81) between the components, it is crucial to make a decision between first- and second-order models. It is worthwhile to investigate the reasons for the substantial disparity in scale correlations between the original study and the present study, which are 0.2 and 0.81, respectively.

In addition to assessing structural validity and reliability, it is imperative to examine alternative kinds of validity, such as convergent validity, as the current evidence is not sufficiently compelling. The Google Forms link supplied for data collection suggests the incorporation of supplementary measuring instruments in the study.

Unfortunately, there are many small typos, language errors, improperly formatted citations, and inconsistencies in the manuscript, which could be corrected by careful reading. e.g., In Table 2, scewness should be written instead of asymmetry, as it is stated in the text; In Table 4, HTML insted of HTMT; citacions: (Hu L.-T. & Bentler P. M., 1999).Regarding the limitations of the research, although 396 surveys have been collected …

I found your text to be both informative and captivating. In my perspective, the inclusion of supplementary notes, analysis, and a thorough examination of the document aforementioned would significantly augment the scholarly merit of the study.

Author Response

Dear Reviewer,

Thank you very much for your informed comments, which helped us so much in improving the manuscript. We appreciated the time you spent doing this and tried our best to address all your comments. We hope that this revised version of the paper reaches the expected standard, worthy of publication in this journal.

A detailed list of answers to your comments and suggestions is reported below.

Many thanks for your time.

Best regards,

Reviewer 3 Report (New Reviewer)

Comments and Suggestions for Authors

Congratulations to the authors for focusing on such an important topic as nutritional literacy. We have some suggestions to improve the manuscript:

1) A more up-to-date definition of nutritional literacy should be included in the introduction, including the section that refers to food literacy. I suggest introducing Krause et al. (2018)

2) Instrument validation is based on translation starting from the language of the original instrument. The authors translate from English to Spanish. But, despite being published in English, isn't the original scale in Dutch? The correct translation is from it. The authors need to address this aspect or, if applicable, clarify that the original was developed in English.

Comments on the Quality of English Language

None

Author Response

Dear Reviewer,

Thank you very much for your informed comments, which helped us so much in improving the manuscript. We appreciated the time you spent doing this and tried our best to address all your comments. We hope that this revised version of the paper reaches the expected standard, worthy of publication in this journal.

A detailed list of answers to your comments and suggestions is reported below.

Many thanks for your time.

Best regards,

Round 2

Reviewer 1 Report (Previous Reviewer 1)

Comments and Suggestions for Authors

Thank you for improving the manuscript and addressing my suggestions.

Author Response

Dear Reviewer

Thank you very much for your comment

Reviewer 3 Report (New Reviewer)

Comments and Suggestions for Authors

The manuscript has improved in quality. The problem of using the scale translated from English continues. Please note that the translation must be from the original instrument into the Spanish spoken in Peru. The authors of the original instrument published it in English, but the instrument was developed in Dutch and adapted for this population. The correct thing to do is to translate the original directly into Spanish. In order not to lose the data already collected, I suggest that the authors carry out this procedure and assess whether there are any relevant changes in relation to the version used. Then you can discuss this aspect in the manuscript.

Author Response

Dear Reviewer

Thank you very much for your comment

But the scale that we have validated, is the original one published in English. 
The authors, before applying the study we have analyzed the original scale, the same that has been developed by the authors (Vrinten et al., 2023), who generated the items following already existing instruments from the English version of the authors (Craig et al., 2003; Fransen et al., 2014; Pelikan et al., 2014; Van et al., 1997). In total the researchers of the original scale stated that the scale would consist of 53 items, of which 32 came from a previous scale, 11 were modified from an existing scale, and 10 were constructed by the authors of the original article. In the expert evaluation, the number of items was reduced to 48, and then the modified items were redacted; after that, the authors conducted cognitive interviews where 20 items were discarded, leaving a total of 28. Finally, the authors of the original scale state that the scale underwent a reduction in the number of items, and after a reliability validation analysis, the scale was composed of 11 items. 
According to the above, the published scale is in English; therefore, we have proceeded to carry out a rigorous validation process and reliability tests. As explained in the manuscript, the scale was subjected to the back-translation process and after that a focus group was conducted, a process that guarantees semantic validity (Krueger, 2000) to ensure the semantic validity of the instrument in the Peruvian context. In addition, to ensure internal consistency, a Cronbach's alpha greater than 0.70 was demonstrated in the item, which indicates an adequate level of reliability (Agbo, 2010; Bagozzi & Yi, 1988).  Thus, the referred process has allowed and demonstrated that the scale is applicable to the Peruvian context. 

References

Agbo, A. A. (2010). Cronbach’s Alpha: Review of Limitations and Associated Recommendations. Journal of Psychology in Africa, 20(2), 233–239. https://doi.org/10.1080/14330237.2010.10820371

Bagozzi, R. P., & Yi, Y. (1988). On the evaluation of structural equation models. Journal of the Academy of Marketing Science, 16(1), 74–94. https://doi.org/10.1007/BF02723327

Craig, C., Marshall, A., Sjostrom, M., Bauman, A., Booth, M., & Ainsworth, B. (2003). International Physical Activity Questionnaire: 12-Country Reliability and Validity. Medicine & Science in Sports & Exercise, 35(8), 1381–1395. https://doi.org/10.1249/01.MSS.0000078924.61453.FB

Fransen, M. P., Leenaars, K. E. F., Rowlands, G., Weiss, B. D., Maat, H. P., & Essink-Bot, M.-L. (2014). International application of health literacy measures: Adaptation and validation of the newest vital sign in The Netherlands. Patient Education and Counseling, 97(3), 403–409. https://doi.org/10.1016/j.pec.2014.08.017

Krueger, R. C. M. (2000). Focus Groups: A practical guide for applied research (3ra edition).

Pelikan, J., Rothlin, F., & Ganahl, K. (2014, November 3). Measuring comprehensive health literacy in general populations: validation of instrument, indices and scales of the HLS-EU study. Proceedings of the 6th Annual Health Literacy Research Conference.

Van, H., Tafforeay, J., Hermans, H., Quateart, P., Schiettecatte, E., & Lebrun, L. (1997). The Belgian health interview survey (Arch Public Health, Vol. 55).

This manuscript is a resubmission of an earlier submission. The following is a list of the peer review reports and author responses from that submission.

Round 1

Reviewer 1 Report

Comments and Suggestions for Authors

 Abstract:

·      Add that the aim is to assess nutritional literacy in Peru.

·      S-NutLit is a tool of two dimensions and not a variable.

·      What is the ideal measurement model used? This needs to be clarified in the abstract.

·      The abstract could be improved and be more informative. Example: start with why nutritional literacy is important, second how it was the methodology performed, and lastly, what were the results of validity and reliability testing.

Introduction:

·      Include the original definition of nutritional literacy, which is the degree individuals have the capacity to obtain, understand, comprehend, and apply nutrition information to eating choices. Reference: https://pubmed.ncbi.nlm.nih.gov/18174098/

·      Authors need to include the literature on Nutrition Literacy relevant to Latin America. The NLit from Gibbs et al. was culturally and linguistically adapted to the Hispanic/Latino population living in the US and was also tested with the same population during pregnancy. The same instrument was also culturally and linguistically adapted to Brazil and Brazilian Portuguese and also applied to a diverse population. Check the further reference: https://pubmed.ncbi.nlm.nih.gov/29164448/ , 

·      Authors need to further introduction on how nutrition literacy is an individual level characteristic that impacts health.

·      Also, I missed discussion on what previous studies that evaluated nutrition literacy found as association regarding health

·      Finally, why there is a need to develop a nutrition literacy scale specific for Peru – dig deeper into how food is a cultural construct.

Materials and Methods:

·      Was the instrument translated back to English to measure how similar the translation was to the original concepts of the instrument?

·      Was the focus group selected based on a convenience sample or the population of interest?

·      How was the number of 6 considered ideal for the focus group for assessment?

·      How was it measured the agreement among individuals reviewing the translation from the instrument? Which statistics were used?

·      What is the Peruvian legal age? 18? 21? Please clarify

·      Marital status results should be reported in results, not in methods.

·      Same for income level results, it should be reported in the results session, not methods.

·      The authors should have included more information about exclusion criteria, as well as how the recruitment was performed. It was clear the surveys were shared through WhatsApp and Telegram, but how were people selected to have surveys sent to them?

·      Also, how was it guaranteed that information shared through WhatsApp and Telegram were safely protected?

Statistical analysis

·       what was considered as references for Cronbach’s alpha?

·      What about descriptive analysis in the statistical analysis? Which ones were performed? Exploratory factor analysis?

·      Kaiser-Meyer-Olkin test?

·      Bartlett’s test?

·      Confirmatory factor analysis?

·      RMSEA?

·      I see a long list of analysis performed in the results, but no discussion of methodology and criteria for interpretation in the statistical analysis section.

Discussion:

·      Even though the authors report that a semantic validation procedure was performed, there was no reference to it in the methods, nor was it reported in the results.

·      Also, there was no reference to the cultural adaptation of the instrument, a crucial step in making it relevant to the Peruvian population.

·      The instrument was also not tested for association with diet habits or health outcomes, for example, diet quality and the presence of chronic diseases. So, it is an extrapolation of results and being overly ambitious to say the instrument could be used to promote a healthy diet and good health. It needs to be tested before having such strong affirmations.

·      The instrument was performed with adults so the discussion on children is inappropriate.

·      Also, evaluating the instrument itself, it measures critical nutrition literacy, which is the ability to advocate for fairness of nutrition and food production; for example, question NL4: if I have doubt about sustainable nutrition, I know where to find information about it. For example, organic vegetables, free-range eggs, fair-trade coffee, etc. This is a difference concept of nutrition literacy than some of the instruments that were used for comparison. What about the population's ability to identify household food measurements or food groups, for example, which are very basic abilities?

Comments on the Quality of English Language

The manuscript uses correct English grammar but needs to be reviewed for academic English writing. The message needs to be direct and assertive.

Author Response

Ref.: Manuscript ID: 2783055

Adaptation and validation of the S- NutLit Scale: To assess nutritional literacy in the Peruvian population.

Dear Reviewer,

Thank you very much for your informed comments, which helped us so much in improving the manuscript. We appreciated the time you spent doing this and tried our best to address all your comments. We hope that this revised version of the paper reaches the expected standard, worthy of publication in this journal.

A detailed list of answers to your comments and suggestions is reported below.

Many thanks for your time.

Best regards,

Reviewer 2 Report

Comments and Suggestions for Authors

Overall, this paper is a very useful and informative contribution to the study of the psychometric properties of the Peruvian version of the instrument measuring nutritional literacy (S-NutLit Scale).

The authors have provided a clear and concise overview of nutritional literacy and its measures. The aims of the study are well-formulated, and the study design and data analysis appear to be appropriate. This paper makes an important contribution to the literature by showing that the questionnaire is a useful measurement tool in other cultures.

My main comments and criticisms of the paper are as follows:

I think that in the introduction part, it would It may be useful to write a few sentences about the results of the psychometric properties of S-NutLit. Now this description is in the Method section (Validation of the S-NutLit instrument), which is not the best solution. Furthermore, it is important to note that the newly developed scale primarily targets young adults, which contradicts the fact that the Peruvian sample includes individuals beyond this age group.

To determine the number of dimensions, it is necessary to conduct a more precise analysis, such as a parallel analysis, and clarify the expectations and theoretical basis. To determine the number of dimensions, several things should be considered.

The description of the statistical analysis does not address such important questions as what rotation and method the factor analysis was made. In the statistical analysis chapter, you can read only what programs were used to make the analyses and the acceptability of Cronbach's alpha.

It would be worthwhile to illustrate the results of the exploratory factor analysis (EFA) in more detail, e.g., the presentation of crossloadings or communalities.

In the case of CFA, it would be worthwhile to present competing models, as the fit indices can be interpreted well in light of this, e.g., one-dimensional structure or a 2-dimensional structure in a second-order or bifactor arrangement. Merely having acceptable fit indices does not support the idea that it does not make a better model. Since the correlation between the factors is very high (r = 0.81), the choice between first- and second-order models is important.

The interpretation of the obtained results is incomplete. For example, there is no explanation as to why item 7 could be included in another factor.

In addition to structural validity and reliability, it would be very important to test some other forms of validity, e.g., convergent validity, because this is not yet very convincing. Based on the provided data collection Google Forms link, I see that other measuring devices were also included in the research.

Unfortunately, there are many small typos, language errors, improperly formatted citations, and inconsistencies in the manuscript, which could be corrected by careful reading. e.g., .79 and.83 (Table 1), NE: Not Specified (Table 1), „This research demonstrates the distribution of the items that make up the S-NutLit metric to measure nutritional literacy, which has a distribution divided into two factors and has high levels of reliability, with a Cronbach's Alpha of 0.906 and a CR of 0.910, indicators that, being higher than 0.7, are qualified as reliable and valid.910, indicators that when being higher than 0.7 are qualified as reliable and valid; besides having an AVE=0.518; this means that more than 50% of the variance in the items corresponding to the questionnaire is related to the construct, supporting with these indicators a high validity of the instrument.”

I was pleased to read your useful and interesting manuscript. In my opinion, the additional notes, analysis mentioned and careful review of the manuscript above would greatly enhance the value of the paper.

Comments on the Quality of English Language

Unfortunately, there are many small typos, language errors, improperly formatted citations, and inconsistencies in the manuscript, which could be corrected by careful reading. e.g., .79 and.83 (Table 1), NE: Not Specified (Table 1), „This research demonstrates the distribution of the items that make up the S-NutLit metric to measure nutritional literacy, which has a distribution divided into two factors and has high levels of reliability, with a Cronbach's Alpha of 0.906 and a CR of 0.910, indicators that, being higher than 0.7, are qualified as reliable and valid.910, indicators that when being higher than 0.7 are qualified as reliable and valid; besides having an AVE=0.518; this means that more than 50% of the variance in the items corresponding to the questionnaire is related to the construct, supporting with these indicators a high validity of the instrument.”

Author Response

(The authors gave the same response as above.)
